# Phase-field Models of Floe Fracture in Sea Ice

Huy Dinh[1], Dimitrios Giannakis[1,2,3], Joanna Slawinska[2], and Georg Stadler[1]

[1]Courant Institute of Mathematical Sciences, New York University, New York, NY, USA
[2]Department of Mathematics, Dartmouth College, Hanover, NH, USA
[3]Department of Physics and Astronomy, Dartmouth College, Hanover, NH, USA

**Correspondence:** Huy Dinh (hd2238@nyu.edu)

**Abstract.** We develop a phase-field model of brittle fracture to model fracture of sea ice floes. Phase fields allow a variational formulation of fracture using an energy functional that combines a linear elastic energy with a term modeling the energetic cost of fracture. We study the fracture strength of ice floes with stochastic thickness variations under boundary forcings or displacements. Our approach models refrozen cracks or other linear ice impurities with stochastic models for thickness profiles. We find that the orientation of thickness variations are an important factor for the strength of ice floes and study the distribution of critical stresses leading to fracture. Potential applications to Discrete Element Method (DEM) simulations and field data from the ICEx 2018 campaign are discussed.

## 1 Introduction

The fracture of sea ice at intermediate scales, $100\,\mathrm{m}$ to $10\,\mathrm{km}$, impacts ice transport on the basin-scale of order hundreds to thousands of kilometers. However, current models do not account for intermediate-scale fractures whose behavior may respond to recent shifts in climate trends. Positive correlation between ice transport and deformation (Lewis and Hutchings, 2019; Rampal et al., 2009), coupled with the loss of thicker multi-year ice (Meier et al., 2014) and falling ice concentrations, suggests that mechanically weaker ice is fracturing faster leading to earlier melt and increased advection. At these scales, kilometer-long fractures form across floes contributing to lead formation and mechanically weaker sea ice with more open water. Advection is enhanced as the weaker ice pack yields to winds. Moreover, darker, open water lowers the effective albedo and increases melt. Incorporating such fracture processes either directly into Discrete Element Methods (DEMs) or effectively into continuum models has been an important, long-standing challenge to improving sea ice predictions (Blockley et al., 2020; Dempsey, 2000; Weiss and Dansereau, 2017).

Modeling intermediate-scale fractures is challenging and most smaller-scale observations are limited to the lab-scale on the order of $10\,\mathrm{m}$ (Timco and Weeks, 2010; Dempsey et al., 2018; Schulson and Duval, 2009; Coon et al., 1998). In particular, available sea ice elastic moduli currently do not account for scale effects and effective parameters due to re-frozen leads and ridges. Previous studies have connected fracture angles and intermediate-scale models (Wilchinsky et al., 2011; Ringeisen et al., 2021; Plante and Tremblay, 2021; Dansereau et al., 2019; Hibler III and Schulson, 2000). One of the first detailed measurements of intermediate-scale deformation were made during the ICEx 2018 field campaign (Parno et al., 2022), yet, simultaneous measurements of forcings from ocean and atmosphere, and initial ice stresses are not available. In response, we

aim to develop a model that can produce large ensembles of fracture events in realistic settings of floe geometry, material parameters, and forcing.

There are multiple candidates for fracture models in sea ice with various strengths and drawbacks. Lu et al. (2015) used an Extended Finite Element Method (XFEM) that simulates fracture modeled by Linear Elastic Fracture Mechanics (LEFM). The formulation of this model and the interpretation of lab-scale measurements of fracture parameters (Dempsey et al., 2018) are systematically based on LEFM. The authors present models that reproduce realistic crack propagation. A drawback of this approach is that a crack tip must be inserted but in general one does not know where a crack nucleates. Particle or Discrete Element Methods (Tuhkuri and Polojärvi (2018); Hopkins and Thorndike (2006); Kulchitsky et al. (2017); Jirásek and Bazant (1995)) model fracture as a process that emerges from the failure of elastic bonds between particles or jointed edges between polygons. They are widely used in modeling ice-structure interactions. Computationally, fracture profiles from these methods are limited by particle or element geometry. In particular, when there is a need to explore fracture profiles over floe geometries, such experiments are computationally prohibitive. Montiel and Squire (2017) model floes as thin elastic discs that fracture at critical stresses. Their fracture profiles are linear and must be parameterized. While this model could be extended to other modes of fracture and floe shapes, the additional step of parameterizing the space of one-dimensional cracks on a two-dimensional floe is challenging. Finally, one could assume fracture is scale-invariant, and apply models from larger scales. There is extensive literature (Rampal et al., 2019; Hutter et al., 2019; Bouchat et al., 2022) to support such an approach. However, those models aggregate intermediate-scale processes that emerge from the lab-scale. In other words, one would need to boldly conjecture that self-similarity of fracture processes holds from thousands of kilometers down to tens of meters.

Phase-field models of fracture (Bourdin et al., 2008) are an appealing choice for modeling floe-scale fractures. They are based on an energetic formation of deformation and fracture. They can directly incorporate lab-scale observations because they are based on Griffith's theory of brittle fracture (Griffith, 1921), the foundation of LEFM. Phase-field regularize cracks and allow them to be described by the variations in a scalar phase field $s$. Phase-field models have been used to explore crack profiles in different geometries, materials and physical settings. In particular, researchers have used phase fields to simulate processes such as fracture nucleation, propagation, branching and fragmentation (Ambati et al., 2015) in a wide range of materials, including concrete, steel and biological tissue. The literature on these models includes an extensive review of numerical implementations (Wu et al., 2020) and extensions to a variety of elasticity constitutive equations and types of failure. We aim to build on these results and on lab-scale ice measurements to model intermediate-scale sea ice fracture.

In this work, we investigate floe-scale fracture with phase-field models. We simulate fracture under boundary force or displacement conditions for a distribution of stochastic ice thickness fields, which model refrozen ice cracks or ice ridges. Our experiments show that the critical forcing at which fracture occurs depends on the geometry of the thickness anomalies and their relation to the forcing to which the floe is subjected. Additionally, we discuss measurement data from the ICEx 2018 expedition in fracture simulations and of incorporating physical fracture in discrete element models.

The outline of the paper is as follows. In section 2, we discuss the phase-field model and numerical implementation. In section 3, we present our experiments on stochastic weaknesses. Finally, in section 4, we discuss future research directions.

## 2   Methods

In this section, we discuss how phase-field models of brittle fracture can be used to model kilometer-scale fractures. We define a phase-field formulation of brittle fracture with linear elasticity in section 2.1. In section 2.2, we review a staggered algorithm to solve the phase-field model formulation when ice floes are subject to displacement or force boundary conditions. We review sea ice parameters from the literature and discuss how we adapt them to our framework, and discuss the implementation of the approach in the open-source Finite Element Method (FEM) library FEniCS (Alnaes et al., 2015) in sections 2.3 and 2.4.

### 2.1   Phase-field model of brittle fracture

We describe ice floe deformation using a linear constitutive relation. The floe geometry is modeled by a two-dimensional domain $\Omega \in \mathbb{R}^2$, on which we consider displacement fields $\boldsymbol{u} = (u_1, u_2)$. We assume that unbroken ice satisfies plane stress linear elasticity equations, i.e., the stress tensor induced by the strain tensor $\boldsymbol{\varepsilon} := (\nabla \boldsymbol{u} + \nabla \boldsymbol{u}^{\mathrm{T}})/2$ is

$$\boldsymbol{\sigma} = [\lambda \mathrm{tr}(\boldsymbol{\varepsilon}) I + 2\mu \boldsymbol{\varepsilon}], \tag{1}$$

where $\lambda$ and $\mu$ are the Lamé parameters, $\mathrm{tr}(\cdot)$ denotes the trace operator, and $I \in \mathbb{R}^{2 \times 2}$ is the identity matrix. The definition (1) results in a corresponding elastic energy density $\Psi(\boldsymbol{u}) : \Omega \to \mathbb{R}$ defined as

$$\Psi(\boldsymbol{u}) := \frac{1}{2} \boldsymbol{\sigma} : \boldsymbol{\varepsilon} = \frac{1}{2} \left[ \lambda (\mathrm{tr}(\boldsymbol{\varepsilon}))^2 + \mu \mathrm{tr}(\boldsymbol{\varepsilon}^2) \right]. \tag{2}$$

Constitutive relations other than (1) can be used, but in this study we focus on a linear rheology and the corresponding well-studied fracture process.

We use a phase-field model of quasi-static brittle fracture developed by Bourdin et al. (2000). Their work builds on Francfort and Marigo (1998) who developed an energetic formulation of Griffith's theory of brittle fracture. Francfort and Marigo extended Griffith's formulation by adding a surface or fracture energy that describes the cost to create a one-dimensional crack set $\Gamma$. In this approach, displacements are allowed to be discontinuous across $\Gamma$. Moreover, the surface energy is the arc length of $\Gamma$ times the fracture toughness $G$, the energetic cost per unit length to fracture material.

Determining a crack set $\Gamma$ that minimizes the total energy is a numerically challenging problem (known as the free-discontinuity problem; Farrell and Maurini, 2017), as the occurrence of fracture is part of the solution and thus a priori unknown, making it difficult to resolve using a numerical discretization. As a remedy, Bourdin et al. (2000) propose an approach that replaces the crack set with a continuous, scalar phase field $s : \Omega \to \mathbb{R}$ describing a diffusive crack profile. The phase field takes values in the interval $[0, 1]$. Fracture occurs where $s = 0$, and ice is unbroken where $s = 1$. Between the two values, $s$ transitions continuously. A length scale parameter $\ell$ controls the spread and smoothness of $s$ near a crack. Bourdin et al. (2000) show that as $\ell \to 0$ in the case of linear elasticity, the zero set of $s$ converges to a minimizing crack $\Gamma$.

Combined with other constitutive relations, phase-field models have also been used to capture different modes of fracture (Ambati et al., 2015). The total energy $\mathcal{E}$ underlying these phase-field models is as follows:

$$\mathcal{E}(\boldsymbol{u}, s) := \underbrace{\int_\Omega (s^2 + \eta)\Psi(\boldsymbol{u})\,dx}_{\text{elastic energy}}$$

$$+ \underbrace{G \int_\Omega \left( \frac{1}{4\ell}(1 - s)^2 + \ell|\nabla s|^2 \right) dx,}_{\text{surface energy}} \tag{3}$$

where $0 < \eta \ll 1$ is a dimensionless residual elasticity parameter, added to prevent loss of a unique minimizing $\boldsymbol{u}$ and ellipticity. Note that $\mathcal{E}$ is convex in $\boldsymbol{u}$ and convex in $s$, but not jointly convex in $(\boldsymbol{u}, s)$. The two terms in (3) represent the elastic energy and the surface energy, and minimizing $\mathcal{E}$ balances these two energy contributions. Points where $s < 1$ reduce the elastic energy, but incur a cost in the surface energy. Such points approximate discontinuous displacements $\boldsymbol{u}$ and thus model full or partial diffusive cracks. If $\ell$ is decreased, the surface energy puts a smaller penalty on $|\nabla s|$ and thus minimizers tend do have thinner crack profiles, i.e., regions where $s$ changes from $s = 1$ to regions where $s \approx 0$ (corresponding to undamaged, and damaged regions and cracks, respectively). Note that in practice, the phase-field function $s$ may also take small negative values or values slightly larger than 1, which does not create any difficulties.

## 2.2 Staggered minimization algorithm

Following Bourdin et al. (2000), we use a staggered algorithm to solve the minimization

$$\min_{\boldsymbol{u} \in U, s \in S} \mathcal{E}(\boldsymbol{u}, s), \tag{4}$$

where $U$ and $S$ are spaces of sufficiently regular functions that satisfy appropriate Dirichlet and Neumann boundary conditions on the boundary $\partial\Omega$. Typical Dirichlet conditions are:

$$\boldsymbol{u} = \boldsymbol{u}_0 \text{ on } \partial\Omega_D \subset \partial\Omega, \tag{5a}$$

$$s = 1 \quad \text{on } \partial\Omega. \tag{5b}$$

In (5a), $\boldsymbol{u}_0$ is a given displacement on part of the boundary $\partial\Omega_D \subset \partial\Omega$. The condition (5b) is the assumption of undamaged material on the boundary and common in phase-field models. On the remaining boundaries, natural (i.e., Neumann) conditions are assumed for $\boldsymbol{u}$.

The algorithm to solve (4) alternates between minimizing with respect to $\boldsymbol{u}$ while keeping $s$ fixed, and minimizing with respect to $s$ while keeping $\boldsymbol{u}$ fixed. Since $\mathcal{E}$ is quadratic in $\boldsymbol{u}$ and $s$, these minimizations can be done exactly. To be more precise, for fixed $s$, the displacement $\boldsymbol{u}$ that minimizes $\mathcal{E}$ must satisfy $\delta_{\boldsymbol{u}}\mathcal{E}(\boldsymbol{u}, s) = 0$, i.e., the variation with respect to $\boldsymbol{u}$

vanishes. Using integration by parts, this results in the linear elasticity equation

$$\nabla \cdot [(s^2 + \eta)\boldsymbol{\sigma}] = \mathbf{0} \quad \text{on } \Omega, \tag{6a}$$

$$\boldsymbol{u} = \boldsymbol{u}_0 \text{ on } \partial\Omega_D, \tag{6b}$$

$$\boldsymbol{\sigma}\boldsymbol{n} = \mathbf{0} \quad \text{on } \partial\Omega \setminus \partial\Omega_D, \tag{6c}$$

where $\boldsymbol{n}$ is the unit normal on the boundary. The condition (6c) results from partial integration.

When $\boldsymbol{u}$ is held fixed, minimizing with respect to the phase fields $s$ shows that the minimizer must satisfy $\delta_{\boldsymbol{s}}\mathcal{E}(\boldsymbol{u}, s) = 0$. Integration by parts shows that this is equivalent to

$$\ell\Delta s - s\left(\frac{1}{4\ell} + \frac{\Psi(\boldsymbol{u})}{G}\right) = -\frac{1}{4\ell} \text{ on } \Omega, \tag{7a}$$

$$s = 1 \quad \text{on } \partial\Omega. \tag{7b}$$

This equation shows that the elastic energy density drives the phase field solution. In particular, if $\Psi(\boldsymbol{u}) \equiv 0$, then $s \equiv 1$, i.e., the ice floe is undamaged. The staggered minimization algorithm is summarized as Algorithm 1. It proceeds by alternately solving (6) for fixed phase field $s$ and solving (7) for fixed displacement $\boldsymbol{u}$. The algorithm terminates either if the maximal update of the phase-field function is less than a given threshold $\epsilon$ or after a maximum number of iterations $N$.

---

**Algorithm 1** Staggered Minimization for Phase-field Model of Fracture

---

   **require** $N > 0$ and $1 \gg \epsilon > 0$

   **set** $n = 1$, $\epsilon^* = 1$ and $s_0 = 1$

   **while** $n < N$ and $\epsilon < \epsilon^*$ **do**

      Solve (6) for $\boldsymbol{u}$

      Solve (7) for $s$

      $\epsilon^* \leftarrow \max_\Omega |s - s_0|$

      $s_0 \leftarrow s$

      $n \leftarrow n + 1$

   **end while**

---

## 2.3 Floe-scale parameters

We introduce the ice thickness $h(x) > 0$ into our model by assuming that the elastic strength is proportional to the ice thickness. Measurements of sea ice elastic moduli are typically not made at the kilometer scale (Timco and Weeks, 2010). In real ice, scale effects such as ice creep come into play as we transition from lab-scale to the size of a floe. In our model, we limit scale effects to those imparted by variations in ice thickness. These variations in thickness lead to variations in elastic strength, which may support concentrating stress. Hence, such variations contribute to the initiation of fracture. For this reason, relative differences in elastic strength are more important than absolute values.

We use reference Lamé parameter values from available sea ice measurements; namely, Young's modulus $E = 9 \times 10^9\,\mathrm{Pa}$ and Poisson's ratio $\nu = 0.3$ (Timco and Weeks, 2010; Schulson and Duval, 2009; Dempsey et al., 1999). The corresponding Lamé parameters vary spatially due to ice impurities or the ice height $h(x)$, i.e., $\lambda$ and $\mu$ are given in terms of $E$, $\nu$ and $h(\cdot)$ as

$$
\begin{aligned}
\lambda(x) &= h(x)\frac{E\nu}{(1+\nu)(1-2\nu)}, \\
\mu(x) &= h(x)\frac{E}{2(1+\nu)}.
\end{aligned}
\tag{8}
$$

For the fracture toughness we choose $G = 10\,\mathrm{J/m^2}$ under the same scale assumptions. Meter-scale measurements of fracture energy (or apparent fracture toughness in the sea ice literature (Dempsey, 1991)) vary between 8 and 12 $\mathrm{J/m^2}$. To our knowledge, no floe-scale measurements of $G$ are available.

## 2.4 Discretization and solvers

Our implementation is based on the open-source finite element software FEniCS (Alnaes et al., 2015). We use linear finite elements on triangular meshes to discretize $\boldsymbol{u}$ and the phase-field function $s$. The scale of fractures occurring in the model depends on $\ell$, and the discretization mesh needs to be fine enough to resolve these fractures. As can be seen in (3), $\ell$ multiplies the norm of the gradient and thus larger $\ell$ results in smaller gradients and thus smoother phase-field functions $s$. The required mesh resolution can be compared to $\ell$ with a one-dimensional solution to (7) under simplified assumptions, as in Kuhn and Müller (2010); Wu et al. (2020). This one-dimensional problem assumes the system is in steady state, neglects elastic energy and models a crack at a single point. We compare the transition between undamaged and damaged state so that it is well-resolved by our mesh points. Mesh sizes are chosen so that the diffusive regions near cracks contain at least 4 or 5 nodes. The matrix systems occurring upon discretization of (6) and (7) are solved using the direct solver MUMPS (Amestoy et al., 2019).

## 3 Fracture behavior of floes with random linear weaknesses

In this section, we use the phase-field formulation described in section 2 to study the effect of varying ice thickness profiles on the fracture behavior of ice floes. First, in section 3.1, we present a stochastic model to insert regions of thinner ice into our models, introducing regions for nucleation of fractures. Then, in sections 3.2 and 3.3, we study the statistical behavior of fractures arising in these models when the ice is subject to boundary displacements or boundary forces.

## 3.1 Experiment setup

For the following experiments, we use a $1\,\mathrm{m}$ thick, $1\,\mathrm{km} \times 1\,\mathrm{km}$ ice floe domain. Other floe geometries could be chosen, but for simplicity we restrict our study to a square domain. Since the occurrence of fracture is inextricably linked to geometry or stress localization induced by material imperfections, we introduce random ice thickness variations into our ice floes. These features model ice impurities and other imperfections, e.g., those arising when an ice floe has gone through a fracture and refreezing cycle. Other possibilities for modeling ice thickness variations include random models of ice height (Bowen et al., 2018), or

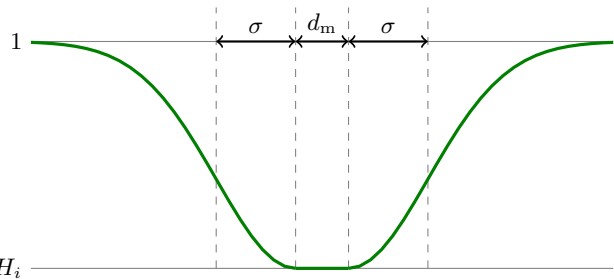

**Figure 1.** Sketch of the mollifying function (9) used to define the ice height $h_i(x)$ in the DFN model.

surface generation methods from random field models such as Gaussian fields (Rasmussen et al., 2006). Here, our focus is on fractures that occur along linear features which are not captured by those methods. We model these features directly with a simpler model.

Our stochastic model is similar to a model of random linear features, also called Discrete Fracture Networks (DFNs) (Min et al., 2004). In these models, a configuration of $K$ line features (here, we fix $K = 10$) is parameterized by centers $c_i$, angles $\theta_i$ and line lengths $b_i$, where $i = 1, \ldots, K$. We extend the DFN model by adding an ice height $H_i$. Angles, centers and heights are drawn from uniform distributions: $\theta_i \in [0, 2\pi], c_i \in \Omega$ and $H_i \in [0, 1]$. Random lengths $b_i$ are uniformly drawn from $[500\,\mathrm{m}, 1500\,\mathrm{m}]$. Each line feature is spatially smoothed using a Gaussian-like mollifier into a height function $h_i(x)$. The mollifying function is displayed in Figure 1. It is parameterized by the minimum height $H_i$, a width $d_m$ and a Gaussian spread parameter $\sigma > 0$, the latter two of which we choose as constants $d_m = 5$ and $\sigma = 5$. For a single line feature on a floe of height $1\,\mathrm{m}$, the height field is given by

$$h_i(\boldsymbol{x}) = 1 - (1 - H_i) \exp\left( -\frac{\xi(\boldsymbol{x})^2}{2\sigma} \right), \tag{9}$$

where

$$\xi(\mathbf{x}) = \max(0, d(\boldsymbol{x}) - d_m).$$

Here, $d(\boldsymbol{x})$ is the distance of a point $\boldsymbol{x}$ to the line segment. This mollification prevents height discontinuities that may be difficult to be resolved in computations and might result in unphysical stresses. We set $h(x) = \min_i h_i(x)$, i.e., the ice thickness is the minimal thickness over the thickness fields generated from the different line segments. The resulting height field has the full height of $1\,\mathrm{m}$ away from line features, is $H_i$ along lines, and takes the minimum height at line intersections. Random realizations of this stochastic model can be seen in the top row in Figure 3.

In our experiments, we assume the following boundary conditions on the right side $\partial\Omega_r$ of $\Omega$, and on the horizontal (i.e., top and bottom) sides $\partial\Omega_h$ of $\Omega$:

$$\boldsymbol{u} = \boldsymbol{0} \quad \text{on } \partial\Omega_r, \tag{10a}$$

$$\boldsymbol{\sigma}\boldsymbol{n} = \boldsymbol{0} \quad \text{on } \partial\Omega_h. \tag{10b}$$

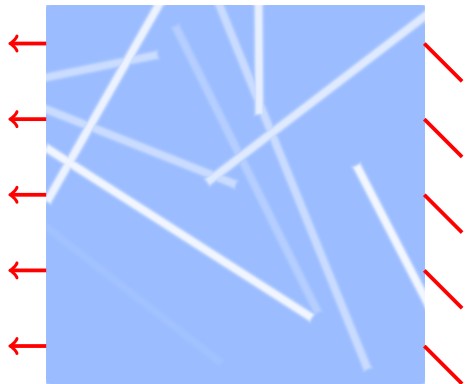

**Figure 2.** Illustration of experimental setup. The floe is fixed along the right edge, stress-free along the top and bottom edges and subject to normal stress or displacement along the left edge. The blue color indicates the floe thickness, where lighter regions are thinner.

That is, the ice floe is held fixed on the right boundary $\partial\Omega_r$ and satisfies stress-free boundary conditions on the horizontal boundaries. On the left side $\partial\Omega_l$, we either prescribe displacements in the normal or tangential direction (see section 3.2), or apply a normal force until fracture occurs (see section 3.3). An illustration of this setup is shown in Figure 2.

In all experiments below, we use $\ell = 1$ and discretize the ice floe domain using $200 \times 200$ squares, each of which is split into two triangles. This results in a resolution of $5\,\mathrm{m}$ and about 5 mesh points to resolve the diffusive phase-field function in direction orthogonal to a fracture. In all experiments in this work, we use the numerical parameters $\epsilon = 10^{-6}$ and $N = 5000$ in Algorithm 1. With these parameters, the staggered algorithm typically terminated after between 1000 and 2000 iterations.

## 3.2 Fracture arising from boundary displacement

We first study fractures that occur as a result of a prescribed tensile displacement. In these numerical experiments, we augment the displacement boundary conditions (10) with the following condition on the left boundary:

$$\boldsymbol{u} \cdot \boldsymbol{n} = -D, \quad \boldsymbol{\sigma}\boldsymbol{n} \cdot \boldsymbol{t} = 0 \ \text{ on } \partial\Omega_l, \tag{11}$$

where we choose $D := 5\times 10^{-3}\mathrm{m}$ and $\boldsymbol{t}$ is the tangential unit vector. That is, we impose displacement Dirichlet conditions in the normal direction and stress-free conditions in the tangential direction. In Figure 3, we show examples of resulting displacement fields and fractures for various samples from our stochastic ice thickness model. We observe that fractures, if they occur, tend to follow linear features of thinner ice. Interestingly, the second crack profile from the left created a closed loop. Whether the ice floes fracture completely, for the given boundary displacement, depends on the height profile. We observe that we are in a regime where complete fractures may or may not occur.

A staggered algorithm or any method that computes critical points of (3) may terminate in local minima or saddle points when boundary conditions are too large and stress localization is absent (Bourdin et al., 2000, 2008; Amor et al., 2009). Our displacement experiments indicate that we are near the threshold of fracture and the inserted weaknesses initiate cracks by

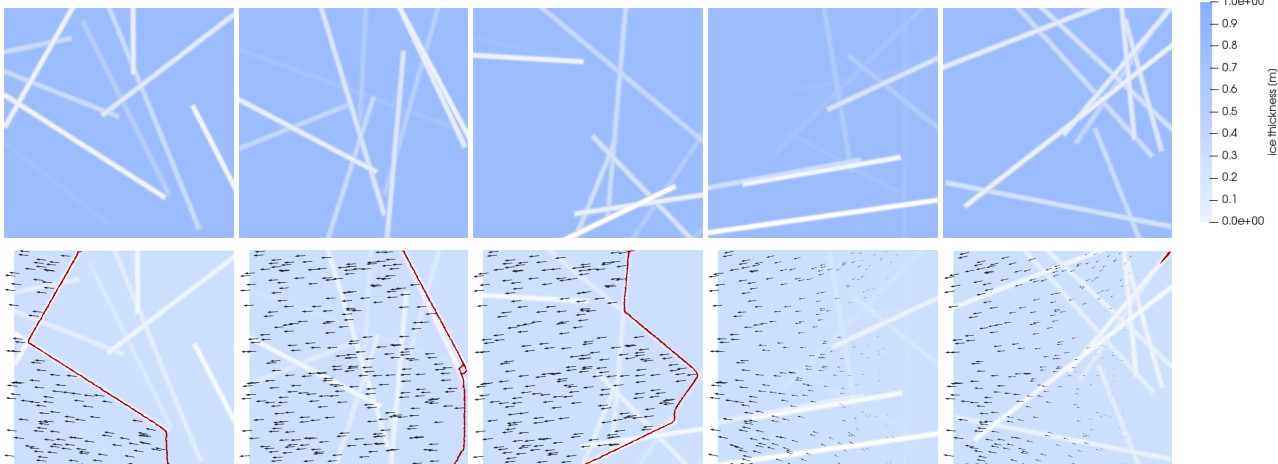

**Figure 3.** Top row: Random realizations of ice thickness; brighter features correspond to thinner ice. Bottom row: Corresponding fractures under tensile stress, where the phase field $s = 0$ (red), may arise due to given displacement of the left boundary in the leftward direction. Black arrows illustrate the displacement field $\boldsymbol{u}$ over the ice floe. Fracture does not occur for all ice thickness fields.

localizing stress. We cannot ensure that Algorithm 1 finds a global minimum but by being close to the critical stress needed for fracture to occur and by introducing inhomogeneities, we avoid two known issues that may lead to spurious minima.

In separate calculations, we have performed compression experiments by enforcing a positive normal displacement in (11). These experiments led to qualitatively similar fracture patterns to those in the tension experiments displayed in Figure 3, so we do not discuss them further.

Next, we consider a follow-up experiment on the ice thickness realizations with shear displacements. Different from (11), we now use a shear displacement condition given by

$$\boldsymbol{u} \cdot \boldsymbol{t} = \bar{D}, \quad \boldsymbol{\sigma}\boldsymbol{n} \cdot \boldsymbol{n} = 0 \ \text{ on } \partial\Omega_l, \tag{12}$$

where $\bar{D} = 5 \times 10^{-2}$. All other boundary conditions are as above. We observe fracture patterns of higher complexity, including fractures into multiple pieces and fractures with multiple sharp turns; see Figure 4 for example fracture profiles and displacement fields. These results demonstrate the ability of the phase-field model to generate fracture with intricate spatial structure under diverse loading and weakness scenarios.

### 3.3    Critical stress distribution

In our next set of experiments, we combine the boundary conditions (10) with a normal pulling force condition on the left edge $\partial\Omega_l$ of the domain, i.e.,

$$\boldsymbol{\sigma}\boldsymbol{n} \cdot \boldsymbol{n} = -L, \quad \boldsymbol{\sigma}\boldsymbol{n} \cdot \boldsymbol{t} = 0 \ \text{ on } \partial\Omega_l. \tag{13}$$

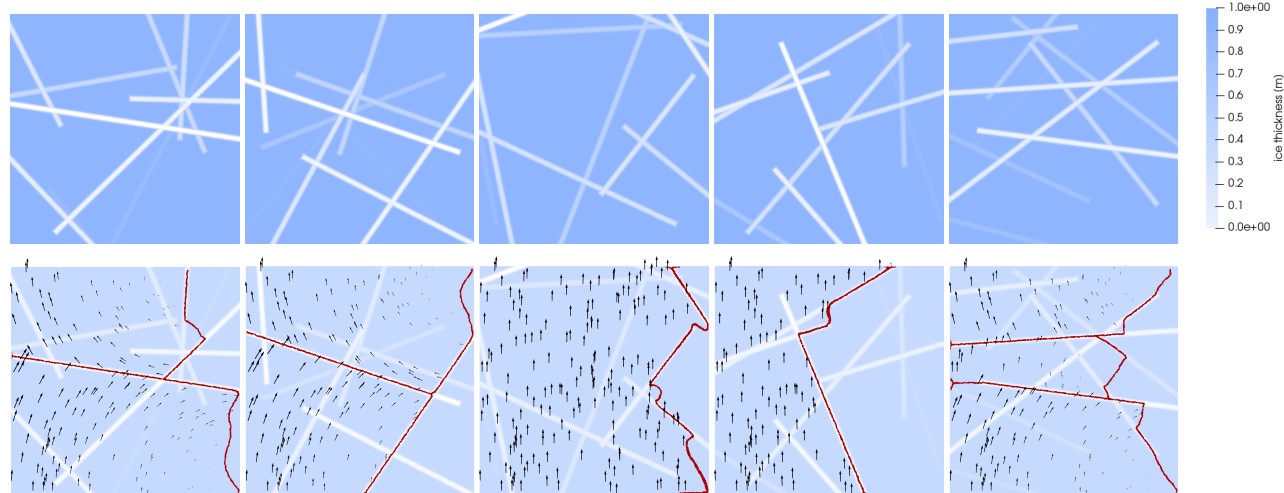

**Figure 4.** Same as Figure 3, but for shear experiments corresponding to the conditions (12) on the left boundary.

We increase the boundary force $L \geq 0$ until the floe completely fractures vertically. Specifically, for each floe with a realization from the stochastic ice thickness model discussed in section 3.1, we increase $L$ in steps of $\Delta L := 5\,\mathrm{kN}$ and minimize the energy (3) using the staggered algorithm. We terminate this incremental loading schedule when we detect complete vertical fracture, and we call the corresponding value of $L$ the critical stress value. To detect complete horizontal fracture, we check if $\min(s) < s^* = 10^{-3}$ and $\|\boldsymbol{u}\|_\infty > 1\,\mathrm{m}$ are satisfied, where $\|\boldsymbol{u}\|_\infty$ is the maximal displacement vector norm. The first condition verifies that fracture occurs at all, while the second condition detects large displacement that occurs when the floe completely breaks vertically as then the broken-off part is not subject to any Dirichlet conditions and is only connected to the clamped, left part of the floe due to the residual stiffness $0 < \eta \ll 1$ in (3).

Using this approach, we study the distribution of critical stresses for the various ice thickness fields. In Figure 5, we show a histogram of the critical stress at which fracture occurs over the distribution of ice thickness fields. As can be seen, all floes fracture at or below $60\,\mathrm{kN}$. Some ice floes fracture at almost an order of magnitude lower normal forces, depending on their thickness anomalies.

In Figure 6(a), we study correlations between the critical normal stress, the average ice thickness and the orientation between the pulling force and the linear features of thinner ice in our stochastic floe model. Note that the average ice thickness ($x$-axis) correlates with the ice strength as can be seen by the larger number of red-shaded dots on the right of Figure 6(a), indicating larger critical stresses. This correlation is also visualized in Figure 6(b), which shows a scatterplot of critical stress vs. average thickness. The correlation is rather weak, with a corresponding correlation coefficient of $0.39$ though it is statistically significant with significance level $p$ equal to 0 to machine precision based on a $t$-test.

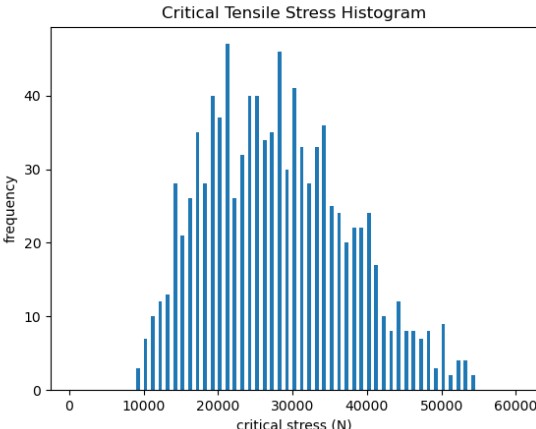

**Figure 5.** Histogram of critical stresses leading to vertical fracture using 1000 random samples of linear weakness fields.

Next, we study if the angle between the force direction and the linear features of thinner ice effects the floe strength. We define the average absolute value of the sine of the angles of the linear thinner ice regions as

$$\alpha := \frac{1}{K} \sum_{i=1}^{K} |\sin(\theta_i)|. \tag{14}$$

Figure 6(c) shows a scatterplot of critical stress vs. $\alpha$. We observe a mild tendency for ice floes to endure a larger critical stress when $\alpha$ is smaller, i.e., the average angle to the force direction is smaller. The reason for this is that the displacement field largely follows the horizontal boundary force, and thus the vertical line features result in stronger stress concentration. The correlation coefficient is found to be $-0.18$, which is statistically significant at level $p = 3 \times 10^{-8}$.

To further study which weaknesses are most important for floe fracture, we compare floes with all $K = 10$ linear features

of thinner ice as described in Section 3.1 with floes that only retain the most vertical feature of thinner ice. In Figure 7, the critical stresses for the floes with all linear features are plotted against the height of the most orthogonal feature as blue-shaded dots. The shading indicates by how much the critical stress increases when the floe only contains the most vertical feature. The fact that lighter colors appear near the front of maximal critical stress at the given height of most the orthogonal feature indicates that the thickness of the most vertical line feature limits the stress the floe can withstand. This holds despite the fact

that the most vertical feature does not generally go through the entire ice floe. We did not find any striking trends when we compared the critical stress against the orientation of the most vertical line or the height of the thinnest crack. Further, the weak correlations in Figure 6 show that the average thickness and $\alpha$ do not contain significant information on critical stress.

Additionally, we study the critical stress for ice floes with a single vertical feature of thinner ice that spans the entire floe and goes through the middle. In Figure 7, these critical stresses are plotted as red curve against the thickness of the feature. We

find that generally the critical stress of floes with the vertical feature of thinner ice is an upper bound to the critical stresses of the floes with all $K = 10$ weaknesses.

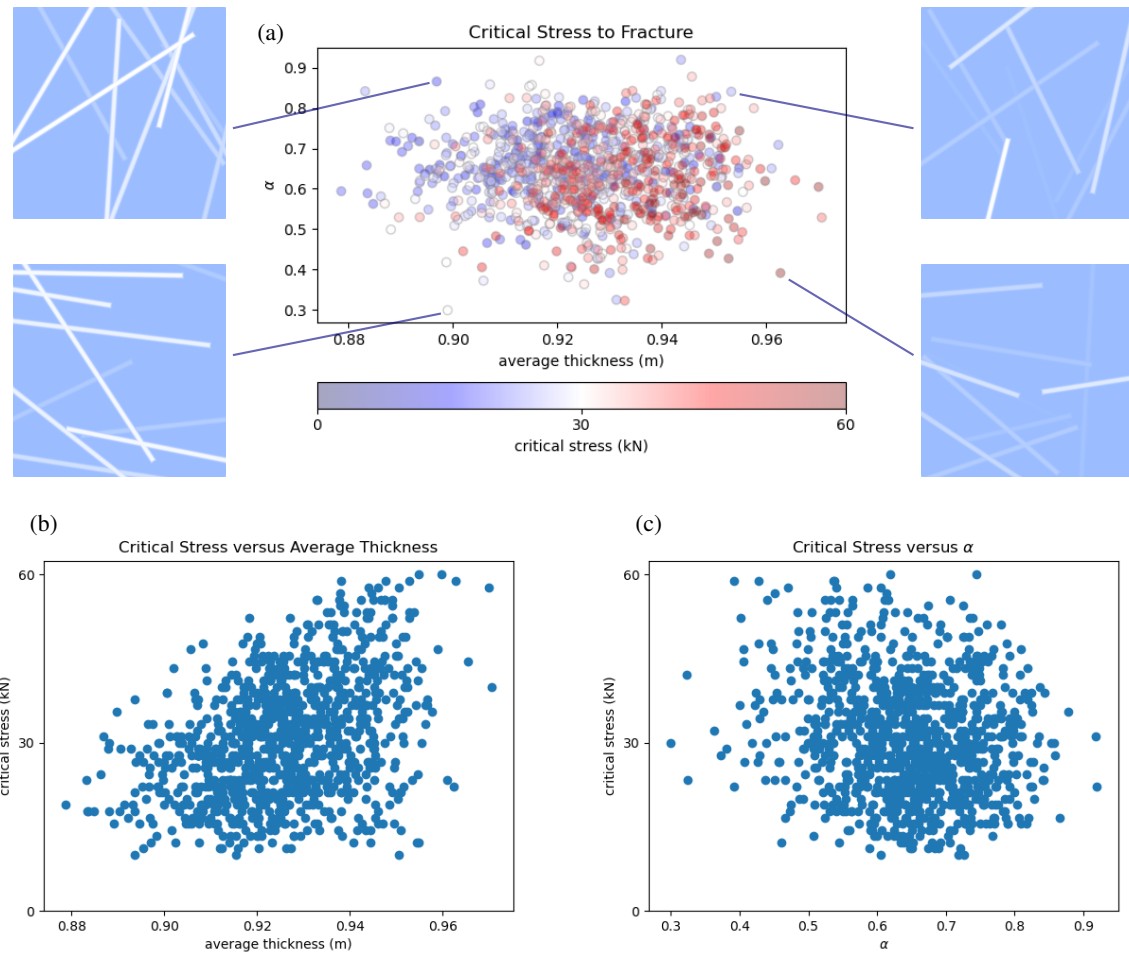

**Figure 6.** Each dot in the scatterplot in Panel (a) corresponds to a random thickness field and the color indicates its critical stress of fracture. Shown on the $x$-axis is the average thickness of each ice floe, and on the $y$-axis a measure for the average orientation of the linear ice features. We also show representative thickness fields corresponding to four of the samples in the scatterplot (top-left and top-right panels). The scatterplots in Panels (b) and (c) show weak correlations, approximately $0.39$ and $-0.18$, between the critical stress, and average thickness and $\alpha$, respectively.

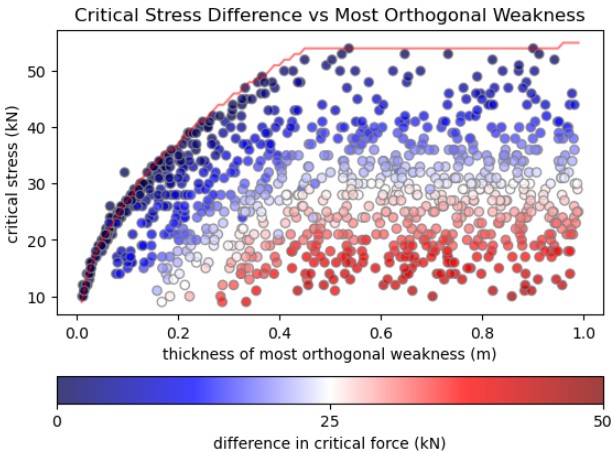

**Figure 7.** Height $H_i$ of the most vertical line ($x$-axis) against the critical stress ($y$-axis). Colors indicate the difference in critical stress when the floe only contains the most vertical feature versus all $K = 10$ features. Dark-blue-shaded dots are random thickness fields where the critical stress between the most vertical line configuration and the full network are largest. The red line indicates the critical stress against the depth of a single vertical spanning line.

## 4    Outlook and challenges

Here, we discuss the potential use of phase-field models for individual floe fractures for incorporation in large-scale DEMs, and for analyzing measurements from the 2018 ICEx field campaign. The following two subsections begin by outlining the applications, followed by discussing technical challenges and potential paths forward.

### 4.1    Fracture in large-scale DEMs

Time-evolving DEM simulations of interacting ice floes are likely to be more accurate if they include realistic intermediate-scale fractures. Ideally, such fracture calculations should not drastically increase computational cost of the time steps in a DEM. This creates critical hurdles to implementing fracture models for floes. Below we discuss potential paths forward to incorporate a physics-based fracture model, such as the phase-field model proposed here, into a DEM.

Existing DEM implementations stimulate sea ice either as tightly bounded particles (Tuhkuri and Polojärvi, 2018; Jirásek and Bazant, 1995) or polygons (Manucharyan and Montemuro, 2022) or polygons joined along edges (Hopkins and Thorndike, 2006). Contact forces between floes or with land masses result in stresses that may lead to fracture. Stress computations can be based on assuming constant stress over an entire floe, or on more detailed floe stress fields computed e.g., using FEM simulations. Based on these collision-caused stresses, a simulation model must decide if a floe fractures (on the floe-scale, fractures are assumed to be instantaneous) and how, i.e., in how many pieces and which directions it fractures.

Computational cost prohibits running the phase-field fracture algorithm in each time step and for each floe. In a typical DEM, each floe element requires estimating stresses due to collisions which, at most, amounts to a PDE solve. Running the

staggered phase-field solver on each element would require many PDE solves for each floe, making the straightforward use of the phase-field fracture model as part of a DEM infeasible. Nevertheless, we imagine two possible approaches to combining phase-field fracture models with a DEM: (1) speeding up the computation using a fracture dictionary generated by a large number of individual ice floe experiments, potentially interpolating between fracture experiments using machine learning; and (2), by detecting the instances when a phase-field fracture computation is crucial to decide whether and how a floe should fracture. In section 3, we produced a sample study of fracture profiles in a fixed geometry and with fixed physical forcing, experimenting with the first approach. However, the occurrence and type of fracture likely depends strongly on floe geometry and involved forces, which would require a massive sample size. In the second approach, we envision that fracture nucleation of a single or multiple elements can be inferred from a large number of experiments. A cheap algorithm could be used to detect when a fracture computation for a floe is necessary in order to initiate the staggered phase-field solver or other algorithm.

A particular challenge would be to model leads across multiple floes, which can be observed in data. Such a behavior could for instance be achieved if one floe's fracture increases the stresses in the neighboring floe, which, as a result, also fractures. This likely requires a fracture model where fractures are initiated through stress localization, and occur in the lead direction. Work in section 3 indicates that the phase-field could use contact forces to produce crack profiles in preferred directions, depending on stresses and ice impurities.

## 4.2 Towards predicting fracture from ICEx 2018 data

Motivated by the high-frequency displacement data available from the ICEx 2018 expedition, we are interested in whether such data can be used to identify fractures when or even before they occur. The ICEx expedition ran from March 8 to 21, 2018, at Ice Camp Skate in Beaufort Sea roughly 230 km north of Prudhoe Bay, Alaska. Researchers spread 24 reflectors on the ice within one kilometer of the camp. The observation area contained both first-year and multi-year ice. A high-precision robotic system measured the location of the reflectors roughly every 1–3 minutes. Measurements were taken both before and after a crack appeared within the observation area. For a more detailed description of the data and its acquisition, we refer the reader to Parno et al. (2022).

There are several challenges to using such data to reproduce and estimate the time and shape of the physical crack. First, using position time series data in a quasi-static fracture model as proposed in this paper is not straightforward. Reflector positions need to be converted to displacements and we do not have information on existing large-scale background stresses or displacements. Second, key data are missing. The experiments on stressed ice floes in section 3 show that boundary stresses and ice impurities largely govern fracture. Thus, the low-resolution ice thickness information, and the fact that only few displacement and even fewer stress observations are available may be limiting. Narrow weaknesses could remain undetected and stress conditions could change sharply.

Parno et al. (2022) estimate boundary displacements from the point displacement measurements under the assumption of a linear elastic model for ice deformation. Using Bayesian inference, they conclude that the linear elastic model is unlikely to explain the observations. They argue that this is due to the occurrence of a fracture that is not captured by the linear elastic mode. One could aim at replacing the linear elastic displacement model assumed in Parno et al. (2022) with the phase-field

fracture model to fit observations. To do so, one could build on theoretical results regarding differentiability of objectives governed by the phase-field model (Neitzel et al., 2017), which is useful to find the best-fitting parameters. However, using the

310 phase-field model to infer the ice state from observations would make the inference problem substantially nonlinear and thus challenging. Another issue, which could also be incorporated in the inference problem, would be to select values for the elastic moduli and fracture toughness from Section 2.3 that are appropriate for the spatial scales of the measurements. We believe that addressing these challenges is an interesting avenue for future work that would allow coupling phase-field models with inference methods to predict sea ice fracture from observational data.

*Code availability.* Our code implementation can be provided by the corresponding author upon request.

*Author contributions.* All authors contributed to the model formulation and the conceptualization of the experiments. HD developed the model code and performed the simulations. All authors contributed to the analysis of the numerical experiments. HD and GS prepared the manuscript with contributions from all co-authors.

*Competing interests.* The authors declare that they have no conflict of interest.

*Acknowledgements.* We appreciate many helpful discussions with Gilles Francfort, Blaise Bourdin, Georgy Manucharyan, Brandon Montemuro and Matthew Parno. This work has been supported by the Multidisciplinary University Research Initiatives (MURI) Program, Office of Naval Research (ONR) grant number N00014-19-1-2421.

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
