# Peer review of "Phase-field Models of Floe Fracture in Sea Ice"

_EGUsphere, 2022_

## Referee Comment (RC1)

**Review of egusphere-2022-790**

**Damien Ringeisen**, *damien.ringeisen@mcgill.ca*

This manuscript presents a new model for the fracture of sea ice at the intermediate scale, from 100 m to 10 km, using the phase-field method. The manuscript describes the model and then presents some tests where the orientation of the inserted fracture set is studied as a function of the critical failure stress. The authors then discuss the challenges of using this model inside a sea ice DEM and the outlook for ice field campaigns.

The manuscript is well-written, concise, clear, and well-presented. It is a valuable contribution to the field of sea ice modeling, especially to the current effort of high-resolution sea ice DEMs. However, I have some comments: I think there is some missing key literature related to sea ice modeling, especially to non-DEM sea ice models and fractures, the relationship between the orientation and the critical stress is a bit unclear, and compressive and shear tests are missing. Finally, I wonder if this manuscript should be better suited for Copernicus' *Geoscientific Model Development (GMD)* journal instead of *The Cryosphere*.

Therefore, I recommend this manuscript for publication with major revisions after my comments have been addressed. Below you will find my general comments, specific comments, and technical corrections. References are listed at the end of this document.

**General comments**

**Research paper or model development paper**

I really like this manuscript; it is a good step for the fracture of floes in DEM models. However, I wonder if it would not be better suited for the *Geoscientific Model Development (GMD)* journal instead of *The Cryosphere (TC)*.

The main result is that the orientation of the lines of reduced thickness is an important factor for fracturing the floe and determines the critical stress. This result is not surprising as embedded lines of reduced thickness reduce the ice strength, so the results look more like a proof-of-concept for the phase-field model for floe fracture than results about sea ice physics. Alternatively, some simulations could be added to describe more the physics of such a model and the behavior of the modeled ice floe and strengthen the manuscript, see my specific comments below.

**Specific comments**

**The notation $\mathbf{u}$ - displacement or velocity?**

There is a confusing notation that needs to be addressed. The authors use the notation $\mathbf{u}$ on L66 for the displacement vector. Then, they define $\mathbf{u}$ as the velocity vector on L140.

Per convention and common use, $\mathbf{u}$ is better suited for velocity. I suggest using something like $\delta x$ for the displacement field and keeping $\mathbf{u}$ for the velocity field. This way, any confusion with the sea ice VP models, where the strain-rate $\dot{\epsilon} = \left( \frac{\partial u_i}{\partial x_j} + \frac{\partial u_j}{\partial x_i} \right)$ with $\mathbf{u}$ the velocity, would be avoided.

**Only tension tests**

This manuscript only shows tensile tests, although observations of sea ice show more compressive or shear deformations. Is there a reason for this choice?

I would like to see how the model behaves with shear and compressive tests, with one example of each. It would strengthen the manuscript to have more examples of how sea ice behaves. I would guess these are forcing situations that are very likely to happen in the ICEx 2018 datasets.

**Missing literature**

I think this manuscript is missing some key literature about sea ice models and observations.

- **Page 1, ca. L20::** I think the author should mention the Elastic Anisotropic Plastic model at this point, which takes into account anisotropy in the ice.
- **Page 2, L38 to L40**: There are many studies studying the self-similarity of sea ice in observations and models. I think those should be cited here. See, for example, Hutter et al. (2019), Bouchat et al. (2022), Rampal (2019)
- **Page 10 and introduction**: The literature linking fracture angles of sea ice to stress in other types of models is missing (e.g., Hibler & Schulson (2000), Dansereau et al. (2019), Plante and Tremblay (2021), Ringeisen et al. (2021), Wilchinsky et al. (2011) )

**relationship between critical stress and orientation of thickness lines**

**Page 10, L210 to 220:**

The correlation between $\alpha$ and *critical force* appears inexistent to me, maybe due to the choice of figure. If the correlation is weak, a correlation coefficient and significance number should be given.

To improve the figure, you could use two panels, one for *critical force* and $\alpha$ and one for *average thickness* and $\alpha$.

The orientation of fractures in sea ice is being investigated in many sea ice models with different physics. It is usually done at larger scales than the ones presented here, but I think it is nevertheless important to mention them. Sea ice rheological models like the VP

or the brittle models (MEB/BBM) set a preferred angle in their physics. I would like to know if this model sets preferred angles (Dansereau et al. (2019), Plante and Tremblay (2021), Ringeisen et al. (2021))).

It would be interesting to see the critical stress dependence with the orientation of a single all-through line of the same reduced thickness, a bit like the study done in Fig. 5., but when the goal is to find the lowest bound of critical stress instead of the highest.

**Numerical cost - advantages compared to other models**

I could imagine doing similar tests with a VP model. However, it would take an enormous amount of computer time because the numerical convergence of the solver is very slow, especially at a 5m resolution. The cost is discussed on page 12, but this model seems much more efficient than the VP, with which I cannot fathom doing 1000s of simulations.

I would be interested to know how fast the model presented here can predict a fracture, e.g., how much time it takes per processor per 1000 random samples.

**Technical corrections**

- **Page 1, L10-12:** I find the sentence unclear
- **Page 3, L58 to L63**: This paragraph is unnecessary. The titles of the subsections are sufficient, and these section introductions are unnecessary.
- **Page 3, L70**: The name of the : operator should be given for clarity.
- **Page 6, L146 to L150**: I think this paragraph is unnecessary (same as above)
- **Page 7, L175:** is it the ice *floe* domain?
- **Page 7, L180**: So the resolution of the experiment is $dx = 5\,\text{m}$. Can you say the number for completeness?
- **Page 11, L232 to L235**: I think this paragraph is unnecessary (same as above)

**Bibliographie**

Bouchat, A., Hutter, N., Chanut, J., Dupont, F., Dukhovskoy, D., Garric, G., Lee, Y. J., Lemieux, J.-F., Lique, C., Losch, M., Maslowski, W., Myers, P. G., Ólason, E., Rampal, P., Rasmussen, T., Talandier, C., Tremblay, B., & Wang, Q. (2022). Sea Ice Rheology Experiment (SIREx): 1. Scaling and Statistical Properties of Sea-Ice Deformation Fields. *Journal of Geophysical Research: Oceans*, *127*(4), e2021JC017667. https://doi.org/10.1029/2021JC017667

Dansereau, V., Démery, V., Berthier, E., Weiss, J., & Ponson, L. (2019). Collective Damage Growth Controls Fault Orientation in Quasibrittle Compressive Failure. *Physical Review Letters*, *122*(8), 085501. https://doi.org/10.1103/PhysRevLett.122.085501

Hibler, W. D., & Schulson, E. M. (2000). On modeling the anisotropic failure and flow of flawed sea ice. *Journal of Geophysical Research: Oceans*, *105*(C7), 17105–17120.

https://doi.org/10.1029/2000JC900045

Hutter, N., Zampieri, L., & Losch, M. (2019). Leads and ridges in Arctic sea ice from RGPS data and a new tracking algorithm. *The Cryosphere*, *13*(2), 627–645. https://doi.org/10.5194/tc-13-627-2019

Plante, M., & Tremblay, L. B. (2021). A generalized stress correction scheme for the Maxwell elasto-brittle rheology: Impact on the fracture angles and deformations. *The Cryosphere*, *15*(12), 5623–5638. https://doi.org/10.5194/tc-15-5623-2021

Rampal, P., Dansereau, V., Olason, E., Bouillon, S., Williams, T., Korosov, A., & Samaké, A. (2019). On the multi-fractal scaling properties of sea ice deformation. *The Cryosphere*, *13*(9), 2457–2474. https://doi.org/10.5194/tc-13-2457-2019

Ringeisen, D., Tremblay, L. B., & Losch, M. (2021). Non-normal flow rules affect fracture angles in sea ice viscous–plastic rheologies. *The Cryosphere*, *15*(6), 2873–2888. https://doi.org/10.5194/tc-15-2873-2021

Wilchinsky, A. V., Feltham, D. L., & Hopkins, M. A. (2011). Modelling the reorientation of sea-ice faults as the wind changes direction. *Annals of Glaciology*, *52*(57), 83–90. https://doi.org/10.3189/172756411795931831

---

## Referee Comment (RC2)

**Review of the article "Phase-field Models of Floe Fracture in Sea Ice" submitted in EGUsphere**

**November 4, 2022**

In this article, authors present the application of a phase-field model, based upon the Griffith fracturation process, developed by B. Bourdin. The fracturation concerns only one square floe fixed on an edge and uniformly tracked on the other. The goal of this study is to develop a tool for the enhancement of simulation performances via a statistical analysis reinforced by observations campaigns.

The phase-field method description is interesting and simulations are performed using the FEniCS library but it is not really an original study. The other key point, random generation of ice scares, is interesting in the context of this article, even if the stochastic model has been adapted from other works. This is the most original part of the article ; thanks to these stochastic generation and an efficient crack generation algorithm, other perform a study of the solidity of floes with specific shape and load. This article article is not very rich in comparisons with experiments, nevertheless, it gives a methodology to generate pertinent simulations. Authors evoked two possible applications, one on DEM, not with direct computation but via a dictionary of behaviors (very pertinent) or via a detection facilitation of the configurations where the phase field computation are needed (I think less pertinent). A remark concerning the simulations, I am quite astonished by the closed line of fracture in Fig. 2, second computation.

As a conclusion, this article is interesting for the good exploitation of complex objects (phase-filed for Griffith modelling of fracturation, stochastic generation of scares) and tracks to exploit the resulting algorithm with a good numerical illustration. I think that this article would be useful for community and deserve to be published.

---

## Referee Comment (RC3)

**Review of "Phase-field Models of Floe Fracture in Sea Ice"**

December 2, 2022

**1 Overview**

Dinh et al. present a phase-field model of sea ice intra-floe fracture. The method is interesting but the analysis of model simulations is cursory. With some additional simulations and analysis this work would be suitable for publication in the Cyrosphere.

**2 Main comments**

- The analysis presented uses a single type of imperfection in the ice pack, namely a fixed number of linear imperfections. Please provide a discussion on why this particular imperfection type was used (i.e.why is it the most appropriate for sea ice) and what other types could be used instead. Also, a brief analysis of the effect on varying the number of imperfections would add to the analysis.

- Figure 3 shows two large peaks significantly higher than the rest of the histogram. Are these statistically significant and if they are what is their cause?

- The analysis in section 3.3 is a quite cursory. I would like to see the analysis presented in figure 4 repeated for other quantities other than just average thickness. It would be interesting to see it repeated with the minimum thickness and with the summed thickness deficit from all the cracks. Also, the average angle of linear ice features used in the analysis (equation 12) should be repeated with weighted averages such as weighted by feature minimum thickness, visible length, or thickness deficit.

- The analysis only considers tension fracture, potentially to avoid complications with ridging in convergence. Some analysis on simulations with shear forcing are warranted though.

**3 Other comments**

- Line 12: "ice area concentrations" → "ice concentrations"

- Line 13: "have impact" → "have had an impact"

- Line 89: "Degeneration near the crack" - describe what this means

- Line 143: Expand briefly the discussion on one dimensional solutions.

- Equation after line 162: It would be useful to see a small diagram of what this crack cross section looks like

- Figure 4: The brightness-based colorbar used for this figure makes the described important features hard to see. I think the figure would be clearer with a color based colormap

- Line 227: This single line paragraph should be merged with another paragraph

- Figure 5: Same comment about colormaps as figure 4.

- Line 283: "theoretical results of Neitzel et al. (2017)" - describe briefly what these are

---

## Author Response (AR1)

**Replies to D. Ringeisen:**

Thanks for your detailed and helpful comments and suggestions. Please find below point-by-point replies (in blue) to your comments and questions (which are reprinted in black). Together with this reply, we also provide an automatically generated diff-file to show all the changes between the originally submitted manuscript and this present version. Please see changes (in red) that follow the replies. Line numbers refer to the revised version.

This manuscript presents a new model for the fracture of sea ice at the intermediate scale, from 100 m to 10 km, using the phase-field method. The manuscript describes the model and then presents some tests where the orientation of the inserted fracture set is studied as a function of the critical failure stress. The authors then discuss the challenges of using this model inside a sea ice DEM and the outlook for ice field campaigns.

The manuscript is well-written, concise, clear, and well-presented. It is a valuable contribution to the field of sea ice modeling, especially to the current effort of high resolution sea ice DEMs. However, I have some comments: I think there is some missing key literature related to sea ice modeling, especially to non-DEM sea ice models and fractures, the relationship between the orientation and the critical stress is a bit unclear, and compressive and shear tests are missing. Finally, I wonder if this manuscript should be better suited for Copernicus' Geoscientific Model Development (GMD) journal instead of The Cryosphere.

Therefore, I recommend this manuscript for publication with major revisions after my comments have been addressed. Below you will find my general comments, specific comments, and technical corrections. References are listed at the end of this document.

**General Comments**

**Research paper or model development paper**

I really like this manuscript; it is a good step for the fracture of floes in DEM models. However, I wonder if it would not be better suited for the Geoscientific Model Development (GMD) journal instead of The Cryosphere (TC).

The main result is that the orientation of the lines of reduced thickness is an important factor for fracturing the floe and determines the critical stress. This result is not surprising as embedded lines of reduced thickness reduce the ice strength, so the results look more like a proof-of-concept for the phase-field model for floe fracture than results about sea ice physics. Alternatively, some simulations could be added to describe more the physics of such a model and the behavior of the modeled ice floe and strengthen the manuscript, see my specific comments below.

**Specific comments**

**The notation u-displacement or velocity?**

There is a confusing notation that needs to be addressed. The authors use the notation on L66 for the displacement vector. Then, they define as the velocity vector on L140. Per convention and common use, is better suited for velocity. I suggest using something like for the displacement field and keeping for the velocity field. This way, any confusion with the sea ice VP models, where the strain-rate with the velocity, would be avoided.

We will clarify the notation when we revise the manuscript. Thanks for pointing this out.

We did not intend to define the vector on L140 (first version) as a velocity. It is a displacement. We have fixed this typo.

**Only tension tests**

This manuscript only shows tensile tests, although observations of sea ice show more compressive or shear deformations. Is there a reason for this choice? I would like to see how the model behaves with shear and compressive tests, with one example of each. It would strengthen the manuscript to have more examples of how sea ice behaves. I would guess these are forcing situations that are very likely to happen in the ICEx 2018 datasets.

We will study shearing and compression experiments when we update the manuscript, and add some of these tests to the paper. If one of the experiments (we believe the shearing experiment is more likely to be promising) proves to be physically interesting, we will perform a statistical study with random ice impurities similar to what we did for the tension test and add it to the manuscript. Thanks for the suggestion—we agree that only tension tests is somewhat restrictive.

We ran simulations of both compressive and shearing displacements and shear forcing. We found the shear displacement produced crack geometry different from compressive experiments. We added a new figure (Fig. 3) with images of the shearing cracks along with discussion in lines 198–204.

**Missing literature**

I think this manuscript is missing some key literature about sea ice models and observations.
Page 1, ca. L20:: I think the author should mention the Elastic Anisotropic Plastic model at this point, which takes into account anisotropy in the ice.

Yes, that reference will make a nice addition to the other intermediate-scale models. We have included a reference to the Elastic Anisotropic Plastic model in line 20.

Page 2, L38 to L40: There are many studies studying the self-similarity of sea ice in observations and models. I think those should be cited here. See, for example, Hutter et al. (2019), Bouchat et al. (2022), Rampal (2019)

We will review and cite these papers. They are more recent than the currently cited papers on self-similarity in sea ice.

We added those references.

Page 10 and introduction: The literature linking fracture angles of sea ice to stress in other types of models is missing (e.g., Hibler & Schulson (2000), Dansereau et al. (2019), Plante and Tremblay (2021), Ringeisen et al. (2021), Wilchinsky et al. (2011) )

Those papers will be added. Thank you for pointing them out.

We referenced these papers in line 20 after revision.

**Relationship between critical stress and orientation of thickness lines**

Page 10, L210 to 220: The correlation between and critical force appears inexistent to me, maybe due to the choice of figure. If the correlation is weak, a correlation coefficient and significance number should be given.

We will add the correlation coefficient and some measure of correlation strength. We do not claim that a strong relation exists and we will emphasize this point in our next update.

We have added the correlations and significance levels of their t-statistics, along with discussion in lines 223-232.

To improve the figure, you could use two panels, one for critical force and and one for average thickness and $\alpha$.

The figure will be improved. We agree that it looks confusing. We have added separate scatter plots and changed to color map of the original figure to improve the contrast.

The orientation of fractures in sea ice is being investigated in many sea ice models with different physics. It is usually done at larger scales than the ones presented here, but I think it is nevertheless important to mention them. Sea ice rheological models like the VP or the brittle models (MEB/BBM) set a preferred angle in their physics. I would like to know if this model sets preferred angles (Dansereau et al. (2019), Plante and Tremblay (2021), Ringeisen et al. (2021))).

This is an interesting question. However, as you point out, these studies consider much larger spatial scales than the individual ice floe scale we consider here. For floe-size scale (i.e., hundreds of meters or a few kilometers), the nucleation and direction of fracture is likely be governed by impurities, by geometric singularities that lead to stress localization, and by concentrated forces due to collisions. Preferred fracture direction can be engineered into the phase field model using anisotropic elastic parameters, but this does not seem desirable for (rotating and moving) ice floes.

It would be interesting to see the critical stress dependence with the orientation of a single all-through line of the same reduced thickness, a bit like the study done in Fig. 5., but when the goal is to find the lowest bound of critical stress instead of the highest.

We will include a study of the floe strength depending of the angle of a single all-through line of reduced thickness. We agree that it's not clear that a vertical line results in the lowest strength for tension experiments.

We ran a simulation and found that the orientation of a single did not have a significant impact on the critical stress. We made no changes.

**Numerical cost - advantages compared to other models**

I could imagine doing similar tests with a VP model. However, it would take an enormous amount of computer time because the numerical convergence of the solver is very slow, especially at a 5m resolution. The cost is discussed on page 12, but this model seems much more efficient than the VP, with which I cannot fathom doing 1000s of simulations. I would be interested to know how fast the model presented here can predict a fracture, e.g., how much time it takes per processor per 1000 random samples.

We appreciate your comments, but believe that such a comparison in favor of the approach presented here is not fair towards VP models. The phase field approach we present is instantaneous, so it does not include time evolution (like VP), and it's also only useful for initially undamaged ice floes. As discussed in the manuscript, if such a phase field model is part of a discrete element model, then in absence of a faster approximate model, many phase field computations for floes in each (or at least many) time steps are necessary, and the time evolution still needs to be taken care of by another model (e.g., the DEM model).

**Technical corrections**

- Page 1, L10-12: I find the sentence unclear
  We will rephrase the sentence. We agree that sentence sounds vague.

  We restructured the sentence for clarity.

- Page 3, L58 to L63: This paragraph is unnecessary. The titles of the subsections are sufficient, and these section introductions are unnecessary.

  We will consider removing it.

  We removed the summary paragraph.

- Page 3, L70: The name of the operator should be given for clarity.
  We will name the operator.

  We have identified the Trace operator.

- Page 6, L146 to L150: I think this paragraph is unnecessary (same as above)
  We will consider removing both.

  We decided to keep this paragraph. We believe it can help readers determine the relevance of the section without reading each subsection.

- Page 7, L175: is it the ice floe domain?
  Yes. We will consider reword the sentence for clarity.

  We clarified by deleting domain.

- Page 7, L180: So the resolution of the experiment is $dx = 5m$. Can you say the number for completeness?
  We added this number to the manuscript.

  We added this information in line 181.

- Page 11, L232 to L235: I think this paragraph is unnecessary (same as above)
  We will consider removing this paragraph.

  We kept the paragraph. Same as above.

**Bibliography**

1. Bouchat, A., Hutter, N., Chanut, J., Dupont, F., Dukhovskoy, D., Garric, G., Lee, Y. J., Lemieux, J.-F., Lique, C., Losch, M., Maslowski, W., Myers, P. G., Olason, E., Rampal, P., Rasmussen, T., Talandier, C., Tremblay, B., & Wang, Q. (2022). Sea Ice Rheology Experiment (SIREx): 1. Scaling and Statistical Properties of Sea-Ice Deformation Fields. Journal of Geophysical Research: Oceans, 127(4), e2021JC017667. https://doi.org/10.1029/2021JC017667

2. Dansereau, V., Demery, V., Berthier, E., Weiss, J., & Ponson, L. (2019). Collective Damage Growth Controls Fault Orientation in Quasibrittle Compressive Failure. Physical Review Letters, 122(8), 085501. https://doi.org/10.1103/PhysRevLett.122.085501

3. Hibler, W. D., & Schulson, E. M. (2000). On modeling the anisotropic failure and flow of flawed sea ice. Journal of Geophysical Research: Oceans, 105(C7), 17105-17120. https://doi.org/10.1029/2000JC900045

4. Hutter, N., Zampieri, L., & Losch, M. (2019). Leads and ridges in Arctic sea ice from RGPS data and a new tracking algorithm. The Cryosphere, 13(2), 627-645. https://doi.org/10.5194/tc-13-627-2019

5. Plante, M., & Tremblay, L. B. (2021). A generalized stress correction scheme for the Maxwell elasto-brittle rheology: Impact on the fracture angles and deformations. The Cryosphere, 15(12), 5623-5638. https://doi.org/10.5194/tc-15-5623-2021 Rampal, P., Dansereau, V., Olason, E., Bouillon, S., Williams, T., Korosov, A., & Samake, A. (2019). On the multi-fractal scaling properties of sea ice deformation. The Cryosphere, 13(9), 2457-2474. https://doi.org/10.5194/tc-13-2457-2019

6. Ringeisen, D., Tremblay, L. B., & Losch, M. (2021). Non-normal flow rules affect fracture angles in sea ice viscous-plastic rheologies. The Cryosphere, 15(6), 2873-2888. https://doi.org/10.5194/tc-15-2873-2021

7. Wilchinsky, A. V., Feltham, D. L., & Hopkins, M. A. (2011). Modelling the reorientation of sea-ice faults as the wind changes direction. Annals of Glaciology, 52(57), 83-90. https://doi.org/10.3189/1727564117959

**Replies to referee #2:**

Thanks for your detailed and helpful comments and suggestions. Please find below point-by-point replies (in blue) to your comments and questions (which are reprinted in black). Together with this reply, we also provide an automatically generated diff-file to show all the changes between the originally submitted manuscript and this present version. Please see changes (in red) that follow the replies. Line numbers refer to the revised version.

In this article, authors present the application of a phase-field model, based upon the Griffith fracturation process, developed by B. Bourdin. The fracturation concerns only one square floe fixed on an edge and uniformly tracked on the other. The goal of this study is to develop a tool for the enhancement of simulation performances via a statistical analysis reinforced by observations campaigns.

The phase-field method description is interesting and simulations are performed using the FEniCS library but it is not really an original study. The other key point, random generation of ice scares, is interesting in the context of this article, even if the stochastic model has been adapted from other works. This is the most original part of the article; thanks to these stochastic generation and an efficient crack generation algorithm, other perform a study of the solidity of floes with specific shape and load. This article article is not very rich in comparisons with experiments, nevertheless, it gives a methodology to generate pertinent simulations. Authors evoked two possible applications, one on DEM, not with direct computation but via a dictionary of behaviors (very pertinent) or via a detection facilitation of the configurations where the phase field computation are needed (I think less pertinent). A remark concerning the simulations, I am quite astonished by the closed line of fracture in Fig. 2, second computation.

We have pointed out the closed line fracture. However, we do not explain why it occurs to avoid speculating without further study.

As a conclusion, this article is interesting for the good exploitation of complex objects (phase-filed for Griffith modelling of fracturation, stochastic generation of scares) and tracks to exploit the resulting algorithm with a good numerical illustration. I think that this article would be useful for community and deserve to be published.

Thank you for your comments. Your evaluation of our work has been helpful in understanding its value.

**Replies to referee #3:**

Thanks for your detailed and helpful comments and suggestions. Please find below point-by-point replies (in blue) to your comments and questions (which are reprinted in black). Together with this reply, we also provide an automatically generated diff-file to show all the changes between the originally submitted manuscript and this present version. Please changes (in red) follow the replies. Line numbers refer to the revised version.

**Overview**

Dinh et al. present a phase-field model of sea ice intra-floe fracture. The method is interesting but the analysis of model simulations is cursory. With some additional simulations and analysis this work would be suitable for publication in the Cyrosphere.

**Main comments**

- The analysis presented uses a single type of imperfection in the ice pack, namely a fixed number of linear imperfections. Please provide a discussion on why this particular imperfection type was used (i.e. why is it the most appropriate for sea ice) and what other types could be used instead. Also, a brief analysis of the effect on varying the number of imperfections would add to the analysis.

  We chose linear imperfections because they are a simple model that spans a range of configurations that lead to different crack profiles. We emphasized that while simple, this choice includes interesting features such as intersections, spanning lines, short lines and networks of weaknesses. Alternative models could include Gaussian random fields e.g., generated with Matern kernels. However, these random models typically generate smooth realizations and thus they might not encourage crack nucleation (and not be more realistic for ice floes that our simple line models). We will explain our preference towards simplicity and discuss potential alternatives in the revised manuscript.

  We have explained our choice of random lines and pointed out alternatives; see lines 153–156.

- Figure 3 shows two large peaks significantly higher than the rest of the histogram. Are these statistically significant and if they are what is their cause?

  These are likely sampling artifacts. We will look into these and update the draft if they persist and thus have physical meaning. Otherwise, we will discuss that the peaks caused by the sampling.

  They were sampling artifacts. We have updated the histogram with more uniformly-spaced bins.

- The analysis in section 3.3 is a quite cursory. I would like to see the analysis presented in figure 4 repeated for other quantities other than just average thickness. It would be interesting to see it repeated with the minimum thickness and with the summed thickness deficit from all the cracks. Also, the average angle of linear ice features used in the analysis (equation 12) should be repeated with weighted averages such as weighted by feature minimum thickness, visible length, or thickness deficit.

  We experimented with other quantities, similar to those suggested, but did not include them in the manuscript as they did not show significant trends. The lack of trends is noteworthy and will be commented on in our other statistical investigations.

  We have provided the other quantities in Lines 239–241.

- The analysis only considers tension fracture, potentially to avoid complications with ridging in convergence. Some analysis on simulations with shear forcing are warranted though.

  We agree that simulations with shear forcing are warranted. As also noted in our response to Damien Ringeisen, we plan to extend the study with a shearing experiment and possibly a compressive one as well.

  We conducted shearing experiments and added their fracture profiles into a new figure. These results are described in lines 198–204.

**Other comments**

- Line 12: "ice area concentrations" → "ice concentrations"

- Line 13: "have impact" → "have had an impact"

- Line 89: "Degeneration near the crack" - describe what this means
  We will correct the above typos.
  These typos have been corrected.

- Line 143: Expand briefly the discussion on one dimensional solutions.
  We will add a short discussion.
  We have added a short discussion in line 138.

- Equation after line 162: It would be useful to see a small diagram of what this crack cross section looks like
  We will add it to the paper or an appendix section. A figure with a plot of the crack cross section has been added.

- Figure 4: The brightness-based colorbar used for this figure makes the described important features hard to see. I think the figure would be clearer with a color based colormap
  We will revise this figure. Thank you for your suggestion.
  We have used a different colormap to improve the contrast.

- Line 227: This single line paragraph should be merged with another paragraph
  We will merge the paragraph.
  It has been merged.

- Figure 5: Same comment about colormaps as figure 4.
  We will adjust the colormap.
  The colormap has been adjusted.

- Line 283: "theoretical results of Neitzel et al. (2017)" - describe briefly what these are
  We will add a description.
  Description has been added.

---

## Referee Report (RR1)

**Review of egusphere-2022-790 - Round 2**

Damien Ringeisen

I thank the authors for their answers to my comments.

I think this manuscript has improved and is almost ready for publication. I recommend this manuscript be accepted with minor revisions, after the authors address my last remaining comments which I list below.

Note:

- *My previous comments are in italic*
- **The author's answers are in bold**
- My new comments are in a normal typeface.

**Comments**

- *Page 3, L58 to L63: This paragraph is unnecessary. The titles of the subsections are sufficient, and these section introductions are unnecessary.*
  - **We will consider removing it.**
  - **We removed the summary paragraph.**
    - There is some confusion here. I think the paragraph you removed is necessary, I was referring to the paragraph between the title of the section (2 Methods) and the title of the subsection (2.1 Phase-field model of brittle fracture.) I think these section introductions are not necessary, but you decided to keep them in the rest of the manuscript, and I accept your choice. However, I think the summary paragraph is necessary, especially the first sentence defining your paper's study. Please add the summary paragraph back.
- *This manuscript only shows tensile tests, although observations of sea ice show more compressive or shear deformations. Is there a reason for this choice? I would like to see how the model behaves with shear and compressive tests, with one example of each. It would strengthen the manuscript to have more examples of how sea ice behaves. I would guess these are forcing situations that are very likely to happen in the ICEx 2018 datasets.*
  - **We will study shearing and compression experiments when we update the manuscript, and add some of these tests to the paper. If one of the experiments (we believe the shearing experiment is more likely to be promising) proves to be physically interesting, we will perform a statistical study with random ice impurities similar to what we did for the tension test**

and add it to the manuscript. Thanks for the suggestion—we agree that only tension tests is somewhat restrictive.

- **We ran simulations of both compressive and shearing displacements and shear forcing. We found the shear displacement produced crack geometry different from compressive experiments. We added a new figure (Fig. 3) with images of the shearing cracks along with discussion in lines 198–204.**
  - I am happy that you (the authors) took the time to perform the additional shear and compression experiments. I find the results of the shear experiments of Fig 3 to be very interesting! However, I do not see a mention of the compression experiments in the manuscript. Is it because you observed the same fracture pattern as in tension experiments? Or do you run into other difficulties in the compression regime?

- *There is a confusing notation that needs to be addressed. The authors use the notation on L66 for the displacement vector. Then, they define as the velocity vector on L140. Per convention and common use, is better suited for velocity. I suggest using something like for the displacement field and keeping for the velocity field. This way, any confusion with the sea ice VP models, where the strain-rate with the velocity, would be avoided.*
  - **We will clarify the notation when we revise the manuscript. Thanks for pointing this out.**
  - **We did not intend to define the vector on L140 (first version) as a velocity. It is a displacement. We have fixed this typo.**
    - I see that $\mathbf{u}$ is used in Griffith fracturation process from Bourdin et al. (2008) as a displacement, I understand that you keep it, but I still find it confusing.

- *To improve the figure, you could use two panels, one for critical force and and one for average thickness and α.*
  - **The figure will be improved. We agree that it looks confusing.**
  - **We have added separate scatter plots and changed to color map of the original figure to improve the contrast.**
    - Thank you for these changes, it is much better.

- *I could imagine doing similar tests with a VP model. However, it would take an enormous amount of computer time because the numerical convergence of the solver is very slow, especially at a 5m resolution. The cost is discussed on page 12, but this model seems much more efficient than the VP, with which I cannot fathom doing 1000s of simulations. I would be interested to know how fast the model presented here can predict a fracture, e.g., how much time it takes per processor per 1000 random samples.*
  - **We appreciate your comments, but believe that such a comparison in favor of the approach presented here is not fair towards VP models. The phase field approach we present is instantaneous, so it does not include time evolution (like VP), and it's also only useful for initially undamaged ice floes.**

**As discussed in the manuscript, if such a phase field model is part of a discrete element model, then in absence of a faster approximate model, many phase field computations for floes in each (or at least many) time steps are necessary, and the time evolution still needs to be taken care of by another model (e.g., the DEM model).**

- Thanks for the precisions, I did not realize that it was so fast, I thought it would still take some time to find the fracture fields as you have iterations in Alg. 1 and describe a solver in Sect. 2.4. What it the usual number of iteration that it takes? You could also give the values used for $\epsilon$ and $N$ of Alg. 1, for reproducibillity.

---

## Author Response (AR2)

**Reply #2 to Dr. Ringeisen:**

Thank you for your additional comments. Please find below point-by-point replies (in blue) to your comments and questions (which are reprinted in black). We only reproduced the unresolved comments and did not keep the entire history of comments but only reprinted what's needed for context.

**Page 3, L58 to L63**

*Referee:* There is some confusion here. I think the paragraph you removed is necessary, I was referring to the paragraph between the title of the section 2 Methods) and the title of the subsection 2.1 Phase-field model of brittle fracture. I think these section introductions are not necessary, but you decided to keep them in the rest of the manuscript, and I accept your choice. However, I think the summary paragraph is necessary, especially the first sentence defining your paper's study. Please add the summary paragraph back.

We apologize for the confusion. We reinstated the last paragraph from the intro. Additionally, we slightly shortened the leading paragraph of Section 2.

*Authors:* We ran simulations of both compressive and shearing displacements and shear forcing. We found the shear displacement produced crack geometry different from compressive experiments. We added a new figure Fig. 3 with images of the shearing cracks along with discussion in lines 198-204.

*Referee:*

I am happy that you (the authors) took the time to perform the additional shear and compression experiments. I find the results of the shear experiments of Fig 3 to be very interesting! However, I do not see a mention of the compression experiments in the manuscript. Is it because you observed the same fracture pattern as in tension experiments? Or do you run into other difficulties in the compression regime?

Yes, the compression experiments produced crack profiles similar to those in the tension experiments. We did not encounter any numerical difficulties in these simulations, and added brief remarks in Sec 3.2 about these compression experiments. Below, in Figure 1 we have provided some pairs of crack and ice thickness profiles.

**Timings for VP vs. phase field models**

*Referee:* Thanks for the precisions, I did not realize that it was so fast, I thought it would still take some time to find the fracture fields as you have iterations in Alg. 1 and describe a solver in Sect. 2.4. What it the usual number of iteration that it takes? You could also give the values used for and of Alg. 1, for reproducibillity.

We have added the values for $\epsilon$ and $N$ to the paper. The number of iterations depends on the experiment. In our case, those with boundary forces should take longer to converge than the displacement experiments. We did not record the number of iterations for the experiments, but based on the results we have, the forcing experiments ran for about 1500 seconds on average. Solves took about one second each, so the average number of iterations is 1500. We added the range of typical iterations to the paper as well.

[Figure]

Figure 1: Random thickness distributions (top row) and corresponding fractures (bottom row). Note that a different color map compared to the paper is used.